# P18: Novel Anticancer Peptide from Induced Tumor-Suppressing Cells Targeting Breast Cancer and Bone Metastasis

**DOI:** 10.3390/cancers16122230

**Published:** 2024-06-15

**Authors:** Changpeng Cui, Qingji Huo, Xue Xiong, Sungsoo Na, Masaru Mitsuda, Kazumasa Minami, Baiyan Li, Hiroki Yokota

**Affiliations:** 1Department of Pharmacology, School of Pharmacy, Harbin Medical University, Harbin 150081, China; cuich@iu.edu (C.C.); qingjihuo1106@gmail.com (Q.H.); xiongxue@iu.edu (X.X.); 2Department of Biomedical Engineering, Indiana University Purdue University Indianapolis, Indianapolis, IN 46202, USA; sungna@iu.edu; 3Frontier Research Institute, Chubu University, Aichi 487-8501, Japan; mitsuda0620@isc.chubu.ac.jp; 4Department of Radiation Oncology, Osaka University Graduate School of Medicine, Suita 565-0871, Japan; k_minami@radonc.med.osaka-u.ac.jp; 5Indiana University Simon Comprehensive Cancer Center, Indianapolis, IN 46202, USA; 6Indiana Center for Musculoskeletal Health, Indiana University School of Medicine, Indianapolis, IN 46202, USA

**Keywords:** breast cancer, anticancer peptides, iTSCs, bone metastasis, GTPase

## Abstract

**Simple Summary:**

This study focused on finding effective anticancer peptides (ACPs) from induced tumor-suppressing cells (iTSCs) to combat breast cancer metastasis to the skeletal system. This study identified P18 as the most potent ACP from a pool of candidates, derived from a protein called Arhgdia. P18 showed significant inhibitory effects on breast cancer cell viability, migration, and invasion, as well as interfering with certain cancer-promoting proteins and GTPase signaling. Importantly, P18 demonstrated enhanced effectiveness when combined with traditional chemotherapy drugs, without significantly affecting healthy cells. Moreover, it mitigated the bone loss associated with breast cancer spread to bones. This study suggests that P18, especially in its modified form (Ac-P18-NH2), could be a promising candidate for breast cancer treatment and preventing bone destruction by regulating specific cellular signaling pathways.

**Abstract:**

Background: The skeletal system is a common site for metastasis from breast cancer. In our prior work, we developed induced tumor-suppressing cells (iTSCs) capable of secreting a set of tumor-suppressing proteins. In this study, we examined the possibility of identifying anticancer peptides (ACPs) from trypsin-digested protein fragments derived from iTSC proteomes. Methods: The efficacy of ACPs was examined using an MTT-based cell viability assay, a Scratch-based motility assay, an EdU-based proliferation assay, and a transwell invasion assay. To evaluate the mechanism of inhibitory action, a fluorescence resonance energy transfer (FRET)-based GTPase activity assay and a molecular docking analysis were conducted. The efficacy of ACPs was also tested using an ex vivo cancer tissue assay and a bone microenvironment assay. Results: Among the 12 ACP candidates, P18 (TDYMVGSYGPR) demonstrated the most effective anticancer activity. P18 was derived from Arhgdia, a Rho GDP dissociation inhibitor alpha, and exhibited inhibitory effects on the viability, migration, and invasion of breast cancer cells. It also hindered the GTPase activity of RhoA and Cdc42 and downregulated the expression of oncoproteins such as Snail and Src. The inhibitory impact of P18 was additive when it was combined with chemotherapeutic drugs such as Cisplatin and Taxol in both breast cancer cells and patient-derived tissues. P18 had no inhibitory effect on mesenchymal stem cells but suppressed the maturation of RANKL-stimulated osteoclasts and mitigated the bone loss associated with breast cancer. Furthermore, the P18 analog modified by N-terminal acetylation and C-terminal amidation (Ac-P18-NH2) exhibited stronger tumor-suppressor effects. Conclusions: This study introduced a unique methodology for selecting an effective ACP from the iTSC secretome. P18 holds promise for the treatment of breast cancer and the prevention of bone destruction by regulating GTPase signaling.

## 1. Introduction

Breast cancer, a malignancy originating in breast tissue, initiates abnormal cell proliferation within breast glandular or ductal tissues, leading to the formation of a cancerous mass [1]. The key risk factors include age, family medical history, genetics, hormone levels, lifestyle choices, and various environmental and physiological elements. Early detection and intervention are imperative for the global prevalence of this disease among women due to the formidable challenge posed by metastasized breast cancer [2]. Treatment modalities include surgical resection, radiotherapy, chemotherapy, hormone therapy, and targeted therapy. Notably, bone metastasis is common, yet there is no established curative treatment strategy [3]. Ongoing research is dedicated to advancing approaches aimed at preventing bone loss and enhancing the survival rates of individuals with bone-metastasized breast cancer [4].

This study aimed to formulate a potent anticancer peptide (ACP) for the treatment of bone metastasis associated with breast cancer. ACPs offer a promising avenue for cancer treatment due to their nonbiological processes in synthesis and quality control, distinguishing them from other cell- and protein-based agents. Typically, composed of basic and hydrophobic residues with positive charges [5], ACPs interact with the negatively charged surfaces of most cancer cells [6]. Although databases like CancerPPD [7], ImmunoPeptidome [8], and PeptideCutter [9] are available for designing ACP candidates and predicting their tumor-binding epitopes [10], effectively designing ACPs remains challenging. The main difficulty lies in identifying specific targets and distinguishing cancer cells from normal cells.

Our novel approach involved deriving ACP candidates from tumor-suppressing proteins secreted by induced tumor-suppressing cells (iTSCs) [11]. Previously, we have demonstrated the generation of iTSCs and their tumor-suppressive conditioned medium (CM) from both cancerous and non-cancerous cells by modulating various signaling pathways. Interestingly, iTSCs can be generated by the overexpression of cMyc or Oct4 [12], unlike induced pluripotent stem cells (iPSCs), which require the overexpression of four transcription factors: cMyc, Oct4, Klf4, and Sox2 [12]. Our earlier research also established that CM derived from mesenchymal stem cells (MSCs) overexpressing Akt and β-catenin selectively inhibits tumor cell proliferation and invasion. Additionally, we found that iTSCs can be induced by activating PKA signaling in Jurkat T lymphocytes or inhibiting AMPK signaling in human peripheral blood mononuclear cells (PBMCs). This raises the intriguing question of whether ACPs can be developed from the secretome of iTSCs.

In our earlier ACP investigations, we highlighted the efficacy of a unique pair, namely, P04 for pancreatic ductal adenocarcinoma and P05 for osteosarcoma (OS) [13,14]. Both P04 and P05 were derived from aldolase A, a glycolytic enzyme known to induce the Warburg effect in cancer cells. In this current study, a preliminary screening of ACPs from a library of fragments of iTSC-derived tumor-suppressing proteins revealed the potent antitumor capabilities of P18 for breast cancer cell lines. P18 is derived from Arhgdia, a Rho GDP dissociation inhibitor alpha. This study aimed to assess the effectiveness of P18 in safeguarding bone from breast cancer cells by exploring the role of GTPase in breast cancer and its associated impact on bone loss.

## 2. Materials and Methods

### 2.1. Cell Culture 

Two human triple-negative breast tumor cell lines, MDA-MB-231 and MDA-MB-436 (ATCC, Manassas, VA, USA), as well as Jurkat lymphocyte cells (ATCC) and peripheral blood mononuclear cells (PBMCs, Lonza, Basel, Switzerland), were cultured in RPMI-1640 (Gibco, Carlsbad, CA, USA). Eight different tumor and bone cell types, cultured in DMEM (Corning Inc., Corning, NY, USA), included MCF7 human estrogen receptor-positive breast cancer cells (ATCC), 4T1.2 mouse mammary tumor cells (obtained from Dr. R. Anderson at Peter MacCallum Cancer Institute, Melbourne, Australia), PANC1 (ATCC) and PANC198 (sourced from Indiana University Simon Comprehensive Cancer Center, Indianapolis, IN, USA) pancreatic cancer cells, xenograft TT2-77 osteosarcoma cells, MG63 osteosarcoma cells (Sigma, St. Louis, MO, USA) [15], RAW264.7 pre-osteoclast cells (ATCC), and murine MSCs derived from the bone marrow of C57BL/6 mice. MC3T3-E1 osteoblasts (Sigma) were grown in αMEM (Gibco, Carlsbad, CA, USA). The culture medium was supplemented with 10% fetal bovine serum and antibiotics (100 units/mL penicillin, and 100 µg/mL streptomycin; Life Technologies, Grand Island, NY, USA). The tumor cells were treated with chemotherapeutic agents, such as Cisplatin (3259, Tocris Bioscience, Minneapolis, MN, USA) and Taxol (3257, Tocris Bioscience), as well as human recombinant ARHGDIA protein (MBS8248528, MyBioSource, San Diego, CA, USA).

### 2.2. Peptides P11 to P22 and P18 Analogs

As ACP candidates, we initially examined 12 peptides (P11 to P22, Genscript Biotech, Piscataway, NJ, USA) (Appendix A) and selected P18 for further analysis in this study. To evaluate whether P18 is located at the effective location along the ARHGDIA protein, we examined P18-N5 and P18-C5, which were shifted along ARHDIA towards the N-terminus and the C-terminus, respectively, by 5 amino acid residues. To further examine the role of specific residues of P18, the eleven P18 analogs, P18-A1 to P18-A11, were evaluated by replacing each of the eleven amino acids of P18 with alanine. Alanine is a non-polar amino acid with uncharged and hydrophobic side chains, which may stabilize P18 but not enhance interactions with binding partners. To improve the efficacy of P18, we further generated three P18 analogs by modifying N- and C-termini with acetyl and amino groups (AC-P18-NH2 and P18-NH2), together with testing a shorter version, P18S, with the same terminal modification (AC-P18S-NH2).

### 2.3. MTT and EdU Assays

Using 96-well plates with approximately 2000 cells per well, an MTT-based metabolic activity assay was performed. The cells were treated with agents and incubated for two days. On the fourth day, they were stained with 0.5 mg/mL thiazolyl blue tetrazolium bromide (M5655, Sigma). The optical density, used to assess metabolic activity, was measured at 562 nm. For the EdU-based proliferation activity assay, the cells were seeded in 96-well plates at about 1000 cells per well. A fluorescence-based cell proliferation kit (Click-iT™, EdU Alexa Fluor™ 488 Imaging Kit; Thermo Fisher, Waltham, MA, USA) was used according to the manufacturer’s instructions.

### 2.4. Scratch Assay

To evaluate the two-dimensional motility of cancer cells, a wound-healing scratch assay was utilized, and the change in the scratch area was determined. On day 1, ~3 × 10^5^ cells were seeded in 12-well plates, and, on day 2, a scratch was made on the cell layer, with the tip of a plastic pipette. Using a light microscope (40×), cell-free zones were imaged after 0 h and 24 h, and the alteration in their areas was quantified by Image J (Version 1.54 f).

### 2.5. Transwell Invasion Assay

The invasion capacity of cancer cells in response to ACPs was assessed using 8 µm pore transwell chambers (353182, Thermo Fisher) in a 12-well plate. The chambers were coated with 300 µL Matrigel (100 µg/mL) and filled with 500 µL of the serum-free medium. After washing the chamber three times with the serum-free medium, approximately 7 × 10^4^ cells in 300 µL serum-free DMEM were placed in the upper chamber, while 800 µL medium was added to the lower chamber. Cells that migrated to the lower side of the membrane were fixed with methanol and stained with crystal violet. Five random images were captured with an inverted optical microscope at 100× magnification, and the average number of stained cells, representing the invasion capacity, was determined.

### 2.6. Western Blot Analysis

Cells were lysed using a RIPA lysis buffer (sc-24948, Santa Cruz Biotech, Dallas, TX, USA) containing protease inhibitors (PIA32963, Thermo Fisher) and phosphatase inhibitors (2006643, Cal-biochem, Billerica, MA, USA). The proteins were separated on 10–15% SDS gels (Bio-Rad Laboratories, Hercules, CA, USA) and transferred to a polyvinylidene difluoride membrane (IPVH00010, Millipore, Billerica, MA, USA). The membrane was incubated with primary antibodies, followed by incubation with secondary antibodies (7074S/7076S, Cell Signaling, Danvers, MA, USA). The antibodies used included those against Snail, p-Src, Src, cleaved caspase 3, caspase 3, RANKL, Cdc42, Col1 and RhoA (Cell Signaling), ALP (Santa Cruz Biotech), OCN (Abcam, Boston, MA, USA), with β-actin (Sigma) as a control. The protein levels were detected using SuperSignal West Femto Maximum Sensitivity Substrate (PI34096, Thermo Fisher) and a luminescent image analyzer (LAS-3000, Fuji Film, Tokyo, Japan) [16].

### 2.7. Fluorescence Resonance Energy Transfer (FRET)-Based GTPase Activity Assay

The three FRET GTPase biosensors were utilized to monitor the activities of Rac1, Cdc42, and RhoA in response to ACPs [17,18]. GTPase activity was estimated based on the emission ratio of yellow fluorescent protein (YFP) to cyan fluorescent protein (CFP), using a Nikon Ti-E inverted microscope with an EMCCD camera (Photometrics, Tucson, AZ, USA), a filter wheel controller (Sutter Instruments, Novato, CA, USA), and a Nikon Perfect Focus System. To reduce photobleaching, an Intensilight fluorescence illuminator (Nikon, Tokyo, Japan) was employed with a neutral-density 32 filter (3% transmittance). The images were captured with a 40× (0.75 numerical aperture) objective, and they were scaled utilizing the color bar, as previously published [19].

### 2.8. Ex Vivo Cancer Tissue Assay

The usage of human triple-negative breast cancer tissue was approved by the Indiana University Institutional Review Board (1911155674), and the tissues were received from the Indiana University Simon Comprehensive Cancer Center Tissue Procurement Core. Freshly isolated breast cancer tissue (~10 mg) was manually fragmented with a scalpel into small pieces (0.5~0.8 mm in length), which were grown in DMEM with 10% FBS and antibiotics for a day. In this assay, we evaluated the tumor-suppressing efficacy of P18, with and without Cisplatin or Taxol. The culture medium, consisting of these agents, was applied for 4 days, and a change in the fragment size was determined.

### 2.9. Bone Microenvironment Assay

Femora were collected from C57BL/6 female mice (~10 weeks old), and the surrounding connective tissues were removed. The femora were cut in half at the mid-diaphysis, and a hole was created at the end of the sample using a 25-gauge needle. 4T1.2 mammary tumor cells (2.5 × 10^5^ cells) were resuspended in 10 μL of culture medium and injected into the bone marrow cavity through the open end of the diaphysis. The bone samples were then placed into a Petri dish and cultured in 1.5 mL of DMEM with 10% FBS and 1% antibiotics at 37 °C and 5% CO_2_. Half of the culture medium was replaced daily with fresh medium. After 2 weeks, the bone samples were harvested for Western blot analysis.

### 2.10. Molecular Docking Analysis

A molecular docking analysis was conducted to evaluate the interactions between the Rho GDP dissociation inhibitor (Arhgdia) and small GTPase (Cdc42), as well as a fragment of Arhgdia, P18, and Cdc42. The structures of Arhgdia and Cdc42 were obtained from the Protein Data Bank PDB database [20], while the P18 structure was assumed as the fragment structure of amino acids 142–152 in the Arhgdia protein. The GRAMM molecular docking software (Vakser Lab-2024, the University of Kansas, KS, USA) [21] was employed to predict the most stable relative configurations of the two interacting molecules. We selected the binding mode and hydrogen bonds with the highest score using the PDBePISA (Protein Data Bank in Europe, Proteins, Interfaces, Structures, and Assemblies) database [22] with the Pymol Educational software (Version 2.3.2) [23].

### 2.11. Statistical Analysis

For the cell-based experiments, three or four independent trials were performed, and the data were expressed as mean ± S.D. Statistical significance was assessed using a one-way analysis of variance (ANOVA). Post hoc statistical comparisons with control groups were conducted using the Bonferroni correction, with statistical significance set at *p* < 0.05. In the figures, the single asterisks indicate *p* < 0.05, and the double asterisks indicate *p* < 0.01.

## 3. Results

### 3.1. Anticancer Effects of P18 on Breast Cancer Cells

In our initial assessment, we screened twelve ACP candidates (P11 to P22) identified from the conditioned medium obtained from MSC-driven iTSCs ([12,17] (Appendix A). These peptides were derived from trypsin-digested fragments originating from a group of proteins within the iTSC CM. Among all the peptides tested, P18 exhibited a significant reduction, on average, in cell viability (Appendix A). Subsequently, we focused on P18 and evaluated its antitumor effects, mostly on breast cancer cell lines and tissues. The decrease in MTT viability of MDA-MB-231, MDA-MB-436, and MCF7 cells was found to be dose-dependent (Figure 1a). The scratch assays demonstrated that P18 effectively reduced the motility of MDA-MB-231 cells (Figure 1b), while the EdU assay and transwell invasion assay revealed that P18 decreased their proliferation and invasive properties (Figure 1c,d). Consistently, the application of P18 to MDA-MB-436 cells inhibited their scratch-based mobility (Figure 1e) as well as their EdU-based proliferation and transwell-based invasive properties (Figure 1f,g).

### 3.2. Antitumor Effect of P18 on Other Tumor Cell Lines but Not MSCs

Our investigation revealed that P18 acted as an ACP not only in breast cancer cells but also in pancreatic ductal adenocarcinoma (PDAC) and osteosarcoma cells. The application of 25 μg/mL of P18 significantly decreased the MTT-based viability of two PDAC cell lines, PANC1 and PAN198 (Figure 2a), as well as the TT2 osteosarcoma cell line (Figure 2b). Moreover, this treatment consistently reduced both the MTT-based viability and scratch-based mobility of 4T1.2 mammary tumor cells (Figure 2c,d). Importantly, however, the same treatment did not exhibit a decrease in the viability or motility of non-tumor cells, such as MSCs, which are precursor cells of bone-forming osteoblasts in the tumor-bone microenvironment (Figure 2e,f). Similarly, the MTT results revealed that P18 did not exhibit any significant effects on the viability of Jurkat lymphocytes, peripheral blood mononuclear cells, and MC3T3 osteoblasts (Appendix A). 

### 3.3. Enhanced Antitumor Effects in Combination with Chemotherapeutic Agents

Given the role of P18 in tumor suppression, we next examined its efficacy with the application of chemotherapeutic agents. The result showed that the application of P18 resulted in a significant reduction in the IC50 value for Cisplatin, from 1.2 µM to 0.5 µM, and for Taxol, from 0.7 µM to 0.3 µM, as determined by the MTT-based viability assay on MDA-MB-231 cells (Figure 3a). 

To further evaluate the efficacy of P18, we exposed it to recently acquired fragments of human triple-negative breast cancer tissue. These tissue fragments were partitioned into two cohorts: one was cultivated in a conventional growth medium, while the other was treated with P18. Throughout four days, we monitored changes in their morphological characteristics. The results demonstrated a significant decrease in the size of ex vivo breast tissue fragments during the four-day P18 treatment (Figure 3b). Moreover, our findings suggested that the concurrent administration of P18 displayed enhanced effectiveness in contrast to the individual use of Cisplatin or Taxol, leading to a more pronounced reduction in the size of cancer tissues after 48 and 96 h treatment (Supplementary Appendix A and Figure 3c). 

### 3.4. Combinatorial Effects with Other ACPs as well as Anti-Osteolytic Effects

We also examined a combined effect with other ACPs. The reduction in MTT viability was observed with other ACPs, such as P04 and P05, and combining these peptides with P18 resulted in an additive effect on the reduction in cell viability (Figure 4a). When applied to MDA-MB-231 and MDA-MB-436 breast cancer cells, P18 downregulated oncoproteins like p-Src and Snail, while simultaneously acting as a cytotoxic agent by increasing the levels of cleaved caspase 3, an apoptotic marker (Figure 4b).

Beyond its impact on breast cancer cell progression, we also investigated the effects of P18 on the development of bone-resorbing osteoclasts. The results revealed that P18, at a concentration of 25 μg/mL, inhibited the differentiation and maturation of RANKL-stimulated osteoclasts. The application of P18 significantly reduced the number of multi-nucleated TRAP-positive osteoclasts (with more than three nuclei) (Figure 4c). Furthermore, the Western blot analysis showed that P18 upregulated the expression of type I collagen (Col1), alkaline phosphatase (ALP), and osteocalcin (OCN), which are involved in osteoblast development (Appendix A).

### 3.5. P18-Containing Protein Arhgdia and Its Binding Partners

As P18 is a fragment of Arhgdia, known as Rho GDP dissociation inhibitor 1, it can bind to Rho GTPases, thereby preventing their binding to effector molecules. This action maintains these molecules in an inactive GDP-bound state, crucially regulating intracellular signaling and various cellular functions like cytoskeletal remodeling, cell migration, differentiation, and proliferation. An extensive analysis of the TCGA database uncovered a correlation between elevated transcript levels of Arhgdia across all cancers and compromised survival rates (Figure 5a). 

As a protein which is enriched in the secretome of iTSCs, Arhgidia acts as a tumor-suppressing protein. Consistently, the use of recombinant human Arhgdia protein (rhArhgdia) resulted in a significant reduction in MTT-based cell viability (Figure 5b). Notably, the IntAct molecular interaction database predicted interactions of Arhgdia with RhoA and Cdc42 (Figure 5c). Additionally, when P18 was applied to MDA-MB-231 and MDA-MB-436 breast cancer cells, a slight upregulation of RhoA and Cdc42 was observed (Figure 5d). This suggests that the effect of P18 involves inhibiting the activity of Cdc42 and RhoA rather than causing the downregulation of their expression.

### 3.6. Effects of P18 on GTPase Activities of Rac1, RhoA, and Cdc42

This study next investigated the role of P18 in regulating the activity of Rho family GTPases. MDA-MB-231 breast cancer cells were transfected with Rac1, Cdc42, or RhoA biosensors and treated with P18 at a concentration of 25 µg/mL for 3 h. The GTPase activity was assessed by measuring the emission ratio of the biosensor within individual live cells. The findings revealed a significant decrease in the activities of RhoA and Cdc42 in response to P18, while no significant change was observed in Rac1 activity (Figure 6a–c). Moreover, between RhoA and Cdc42, the activity of Cdc42 showed a more pronounced decrease compared to RhoA, suggesting that the inhibition of Cdc42 activation may hold greater significance for P18 as an ACP.

### 3.7. Interaction between Arhgdia and Cdc42

The IntAct molecular interaction database highlighted a potential interaction between Arhgdia and Cdc42. To evaluate this connection, protein–protein docking was performed, resulting in a score of 478, suggesting probable binding between Arhgdia and Cdc42. This analysis revealed 14 possible interactions (hydrogen bond distances ranging from 2.1 to 3.7 Å) (Appendix A), with specific amino acid positions, such as GLU 87-LYS 131 and ARG 152-GLU 91, showing hydrogen bonding. Additionally, our investigation demonstrated the probable binding of P18 to Cdc42, unveiling seven putative hydrogen bonds in the range of 2.7 to 3.7 Å (Figure 7a,b, Appendix A).

Next, we aimed to improve the antitumor activity of P18 by shifting the fragment location along the Arhgdia protein, altering amino acid residues, or modifying N- and C-termini. Initially, we sought to ascertain whether P18 represented the most effective segment within the Arhgdia protein. To explore this, we modified the P18 position by shifting the overall sequence by five amino acids towards both the N- and C-termini, resulting in P18-N5 and P18-C5 variants. Metabolic assays based on MTT confirmed that the original P18 retained the strongest tumor-suppressive effect among the four cancer cell lines tested (Appendix A). To pinpoint which specific amino acid within P18 played a pivotal role, we substituted alanine with each of the eleven amino acids present in P18, generating eleven distinct P18 analogs. Subsequent MTT-based metabolic assays reaffirmed that the original P18 exhibited the most potent tumor-suppressive effect in MDA-MB-231 and MDA-MB-436 breast cancer cell lines, as well as in TT2 OS and PANC1 PDAC cells (Appendix A). 

MTT-based metabolic assays showed that P18 modified by N-terminal acetylation and C-terminal amidation (Ac-P18-NH2) exhibited stronger tumor-suppressor effects than the original P18 on three breast cancer cell lines (Figure 7c,d). Similarly, this P18 analog exhibited stronger tumor-suppressor effects than the original P18 on 4T1.2 mammary tumor cells, TT2 OS cells, as well as PANC1 and PAN198 PDAC cell lines (Appendix A). Notably, the MTT-based metabolic assays showed that the C-terminal-amidated P18 alone did not improve the tumor-suppressor effect (Appendix A). The illustration of the P18 analogs is shown in Figure 7e.

### 3.8. Reduction in Oncoproteins by P18 in Ex Vivo Bone Culture

Considering the anticancer effects of P18 observed in breast cancer cells, we further explored its tumor-suppressive potential using ex vivo bone cultures. Illustrated in the operational schematic, femurs inoculated with 4T1.2 mammary tumor cells were cultured for 2 weeks in the presence and absence of 25 μg/mL P18, after which they were assessed for oncoprotein expression. Upon administration of P18, the results revealed a significant reduction in the levels of p-Src in oncogenic signaling, Snail in EMT stimulation, RhoA as GTPase, p-Akt in PI3K signaling, and RANKL in osteoclast development. Conversely, an upregulation was observed in the levels of c-Cas 3, Cdc42, and alkaline phosphatase (ALP) (Figure 8a). A proposed mechanism outlining the action of P18 is depicted in Figure 8b.

## 4. Discussion

In this study, we demonstrated that P18 inhibited the progression of cancer cell lines, including breast cancer, pancreatic cancer, and osteosarcoma, as well as breast cancer tissue fragments, without significantly affecting MSCs. The combination of P18 with chemotherapeutic agents such as Cisplatin and Taxol dramatically reduced the drug’s half-inhibitory concentration (IC50), as well as the combination with other ACPs such as P04 and P05. The Western blot results showed that the application of P18 decreased the expression of proto-oncoproteins while increasing the expression of the apoptosis-associated protein c-Cas 3 in the MDA-MB-231 and MDA-MB-436 breast cancer cell lines. P18 inhibited the GTPase activity of RhoA and Cdc42 without reducing their expression. Consistently, the administration of P18 to mouse femora, inoculated with 4T1.2 mammary tumor cells, significantly suppressed tumor progression. Furthermore, the molecular docking analysis predicted the binding of Arhgdia and P18 to Cdc42, and FRET-based live-cell imaging revealed that P18 greatly reduced the activities of Cdc42 and RhoA. ACPs, particularly in the α-helical form, can penetrate plasma, nuclear, and/or mitochondrial membranes and exert pharmacological activity [24,25]. Predicted to be an α-helical peptide with a molecular weight of 1245 Da, P18 is expected to passively diffuse through cell membranes without endocytosis or specific receptors.

Many chemotherapeutic medications, including DNA-toxic drugs and molecularly targeted antitumor therapies, exhibit myelosuppressive effects [26]. Therefore, it is imperative to minimize anticancer drug doses while maintaining therapeutic efficacy. The combination of P18 with two widely used chemotherapeutic agents, Cisplatin and Taxol, effectively reduced the IC_50_ values. The subsequent application of human triple-negative breast cancer tissue fragments demonstrated that the concurrent administration of P18 was more effective than using Cisplatin or Taxol alone. Moreover, co-culturing the mammary tumor cell line 4T1.2 with mouse femora revealed that P18 significantly downregulated the generation of oncoproteins and RANKL while upregulating the alkaline phosphatase that is linked to bone formation. The result suggests that P18 possesses osteoprotective properties in addition to its tumor-suppressing capabilities.

The iTSC-driven ACP candidates explored in this study set themselves apart from those in other ACP investigations due to their unique origin—peptide fragments derived from tumor-suppressing proteins. Recent advancements in ACP development have focused on enhancing targeting and selectivity. Despite these strides, a majority of ACP clinical trials listed in the U.S. NIH Clinical Trials database are associated with peptide-based vaccines [27]. Notably, vaccines such as P53 and K-RAS target the detection and elimination of malignant cells harboring mutations in p53 and K-Ras [28]. While the side effect of any ACP is a concern, we anticipate a unique interplay of P18 with immune responses, given that P18 originates from tumor-suppressing proteins produced by iTSCs. 

In contrast to our previous examinations of ACPs P04 and P05, both derived from Aldoa, a glycolytic enzyme, P18 presents distinctive advantages and a more comprehensive research profile. Despite the challenges associated with predicting effective ACPs, P18 shows promise as a more potent and efficient ACP when compared with P04 and P05. Notably, P18 demonstrates a stronger inhibitory effect on PDAC and OS cell lines than its counterparts P04 and P05. Furthermore, we conducted a thorough investigation into P18’s anticancer effects, including alanine substitution and two-terminal modifications. The absence of hotspots for anticancer activity in alanine substitution suggests that the overall molecular structure of P18 contributes to its anticancer properties. In acetylation and amidation experiments, P18 modified with N-terminal acetylation and C-terminal amidation (Ac-P18-NH2) exhibited enhanced anticancer activity across multiple cancer cell lines. However, the sole amidation of the C-terminus did not improve the antitumor efficacy. We postulate that N-terminal amidation disrupts intramolecular interactions of P18, thereby increasing its affinity to its target. Interestingly, P18S, a peptide composed of merely seven amino acids, demonstrated anticancer activity comparable to or surpassing that of P18. The high industrial productivity and anticipated high bioavailability of short-chain peptides enhance the potential development of P18S as an anticancer peptide pharmaceutical. This outcome significantly bolsters the prospects for the advancement of anticancer peptide drugs derived from P18S.

Our research is constrained by the absence of in vivo trials, limiting our ability to identify toxicity and half-life. We recognize the significance of in vivo investigations in evaluating drug toxicity and pharmacokinetics, along with the need for the precise assessment of critical parameters. While this cellular experiment provides valuable insights into the potential adjuvant role of ACPs, it has limitations. In vivo, variations in bioavailability and drug metabolism can occur between chemotherapeutic agents and ACPs. Cellular studies lack the complexity of whole organisms, and further preclinical experiments are necessary to validate these findings. Nonetheless, our study remains significant since in vitro tests were conducted to thoroughly examine the mechanism of action, biological effects, and interactions of ACPs with cancer cells. This early exploration provided valuable insights into the potential therapeutic effects of ACPs. Furthermore, our findings contribute to understanding the design, manufacturing, and in vitro assessment of ACPs. To address this research gap, future endeavors may prioritize including in vivo investigations to assess the toxicity levels and half-life of ACPs, as well as refining models to better predict their activity in the organism. Clinically, Thymosin α1, an ACP consisting of 28 amino acids and with a molecular mass of 3018 Da, is administered via subcutaneous injections at a dosage of 1.6 mg twice a week as part of adjuvant treatment [29,30]. For the further evaluation of P18, this regimen may serve as a benchmark for comparing the stability, permeability, efficacy, safety, and side effects of other ACPs in clinical applications.

The exploration of iTSCs-derived ACPs presents a promising avenue for developing therapeutic approaches focused on direct tumor suppression. While substantial progress has been achieved, our future research endeavors will prioritize enhancing the efficacy of these ACPs. The key focus areas for improvement include refining tumor targeting and ensuring safety. ACPs are designed to bind specifically to tumor cells, facilitating direct tumor inhibition while minimizing adverse effects on normal tissues. This targeted approach positions ACPs as potent tools for individualized therapy, adaptable to the specificity of different patients’ tumors. The screening of iTSCs CM-derived ACPs with direct tumor-targeting capabilities holds immense promise in the realm of cancer therapy.

## 5. Conclusions

This study highlights P18 as a promising candidate for inhibiting cancer progression, showcasing its efficacy in suppressing the viability, motility, proliferation, and invasion capabilities of various cancer cell lines, particularly breast cancer cell lines and tissues. Furthermore, P18 demonstrates the inhibition of osteoclast maturation and potential synergistic effects when combined with existing chemotherapeutic agents, reducing the IC_50_ of Cisplatin and Taxol. Mechanistically, P18 inhibits the GTPase function of RhoA and Cdc42 without significantly altering their expression levels. These findings underscore the therapeutic potential of P18 in combating cancer progression and bone metastasis and warrant further exploration of its clinical applications.

## Figures and Tables

**Figure 1 cancers-16-02230-f001:**
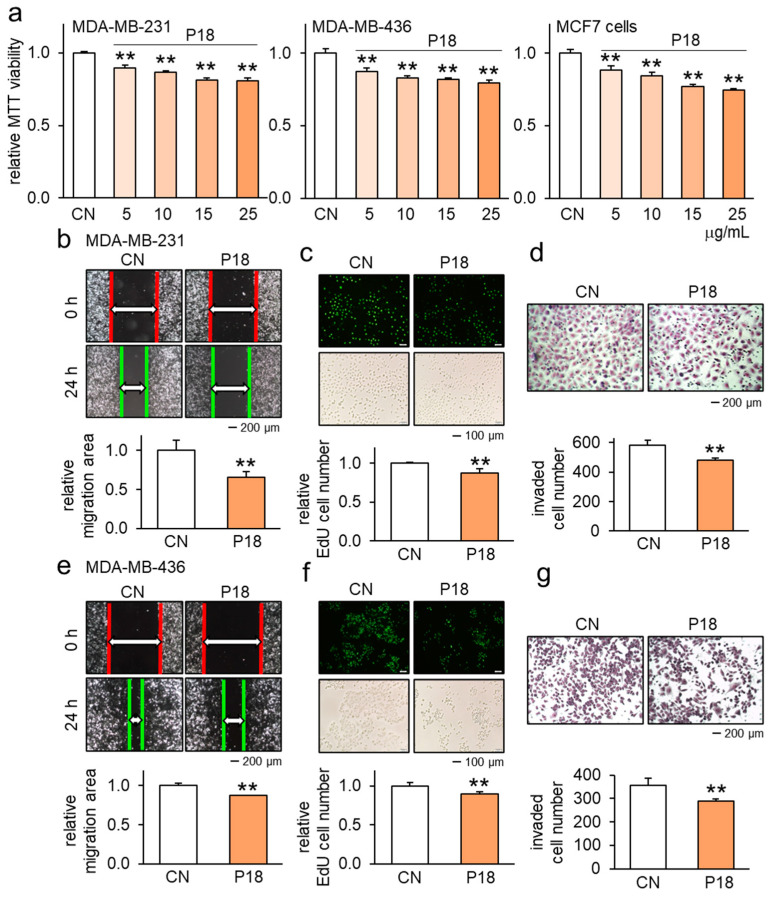
Antitumor effect of P18 on breast cancer cells. CN = control. The double asterisks indicate *p* < 0.01. (**a**) Reduction in MTT-based viability of MDA-MB-231, MDA-MB-436, and MCF7 breast cancer cells in response to 5, 10, 15, and 25 µg/mL of P18. (**b**) Reduction in scratch-based motility of MDA-MB-231 cells in response to 25 µg/mL of P18. (**c**,**d**) Inhibitory effects of 25 µg/mL of P18 on EdU-based proliferation and transwell invasion of MDA-MB-231 cells. (**e**) Reduction in scratch-based motility of MDA-MB-436 cells in response to 25 µg/mL of P18. (**f**,**g**) Inhibitory effects of 25 µg/mL of P18 on EdU-based proliferation and transwell invasion of MDA-MB-436 cells.

**Figure 2 cancers-16-02230-f002:**
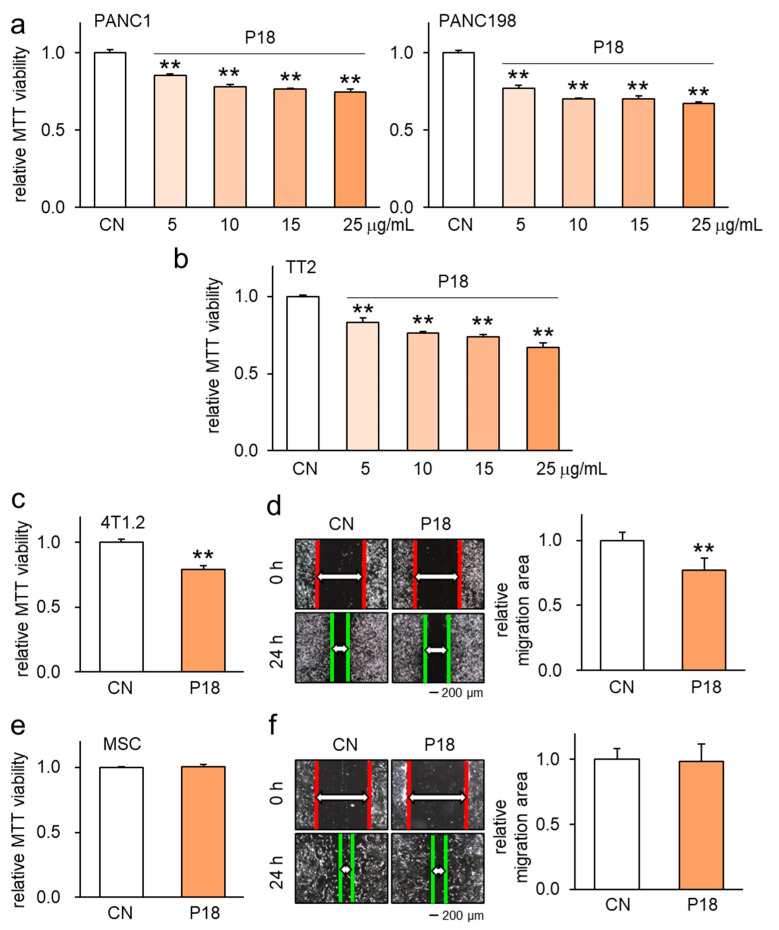
Inhibitory effects of P18 on pancreatic ductal adenocarcinoma (PDAC), osteosarcoma, and breast tumor cells. CN = control. The double asterisks indicate *p* < 0.01. (**a**) Reduction in MTT-based viability of PANC1 and PANC198 PDAC cells in response to 5, 10, 15, and 25 µg/mL of P18. (**b**) Reduction in MTT-based viability of TT2 OS cells in response to 5, 10, 15, and 25 µg/mL of P18. (**c**,**d**) Inhibitory effects of 25 µg/mL of P18 on MTT-based viability and scratch-based motility of 4T1.2 mammary tumor cells. (**e**,**f**) No significant effects on MTT-based viability and scratch-based motility of MSCs in response to 25 µg/mL of P18.

**Figure 3 cancers-16-02230-f003:**
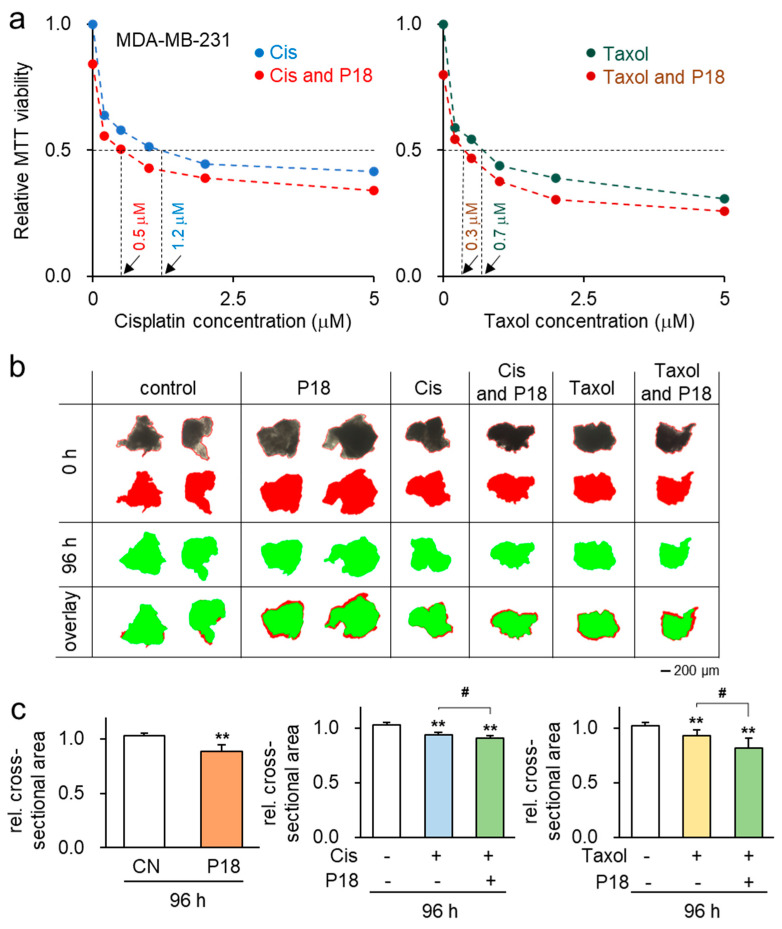
Inhibitory effects of P18 on MDA-MB-231 breast cancer cells and breast cancer tissue fragments in combination with Cisplatin and Taxol. CN = control; Cis = Cisplatin. The double asterisks indicate *p* < 0.01. The single hash marks indicate *p* < 0.05 for the comparison between the Cisplatin and Taxol treatments. (**a**) Additive antitumor effects of P18 with Cisplatin or Taxol. (**b**) Reduction in breast cancer tissue fragment size over 96 h in response to 25 µg/mL of P18 (*n* = 6). The red image shows tissue fragments after 0 h, while the green image shows them after 96 h. (**c**) Reduction in breast cancer fragment size over 96 h in response to 5 µM of Cisplatin or Taxol with and without 25 µg/mL of P18 (*n* = 6). The red image shows tissue fragments after 0 h, while the green image shows them after 96 h.

**Figure 4 cancers-16-02230-f004:**
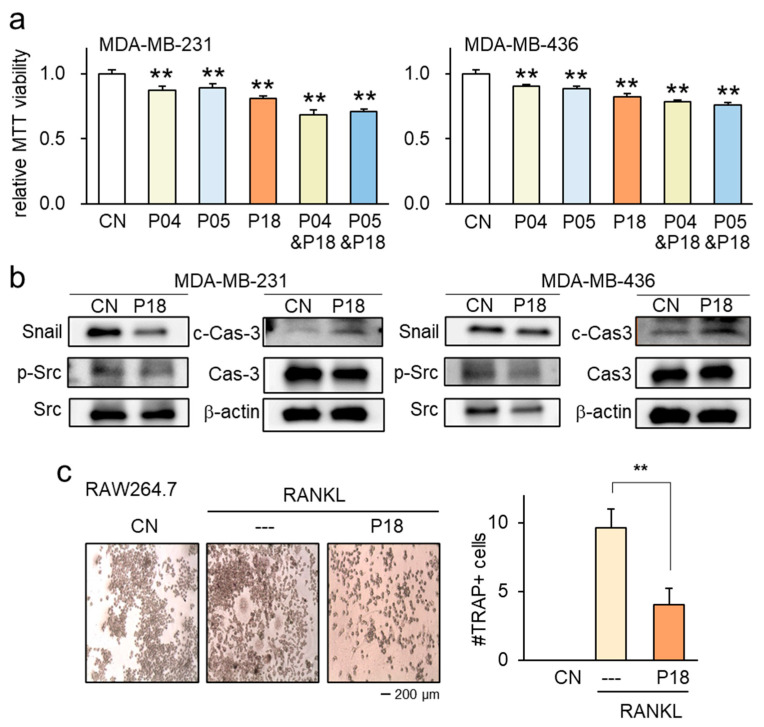
Antitumor effects of P04, P05, P14, and P20 with the combination of P18 on MDA-MB-231 and MDA-MB-436 breast cancer cells. CN = control. The double asterisk indicates *p* < 0.01. (**a**) Effects of P04, P05, P18, and their combinatorial usage on MTT-based viability of MDA-MB-231 and MDA-MB-436 breast cancer cells. (**b**) Decrease in the levels of p-Src and Snail, as well as an increase in cleaved caspase 3 (c-Cas-3) in MDA-MB-231 and MDA-MB-436 breast cancer cells by the application of 25 µg/mL P18. The uncropped blots are shown in Appendix A. (**c**) Significant reduction in multi-nucleated RANKL-stimulated osteoclasts in response to P18.

**Figure 5 cancers-16-02230-f005:**
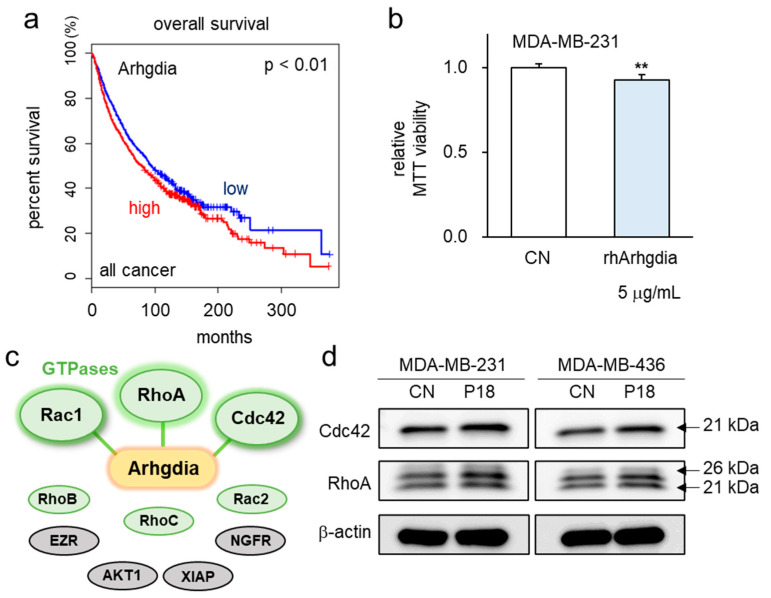
P18 and P18-containing ARHGDIA with their possible binding partners. (**a**) The lowered survival rate for patients with a high ARHGDIA transcript level in the TCGA database (**b**) Reduction in MTT-based viability of MDA-MB-231 cells in response to 5 μg/mL of recombinant ARHGDIA protein. The double asterisk indicates *p* < 0.01. (**c**) Prediction of ARHGDIA-interacting proteins including GTPases such as Rac1, RhoA, and Cdc42. (**d**) Slight increase in the levels of Cdc42 and RhoA in MDA-MB-231 and MDA-MB-436 breast cancer cells by the application of 25 µg/mL P18. The uncropped blots are shown in Appendix A.

**Figure 6 cancers-16-02230-f006:**
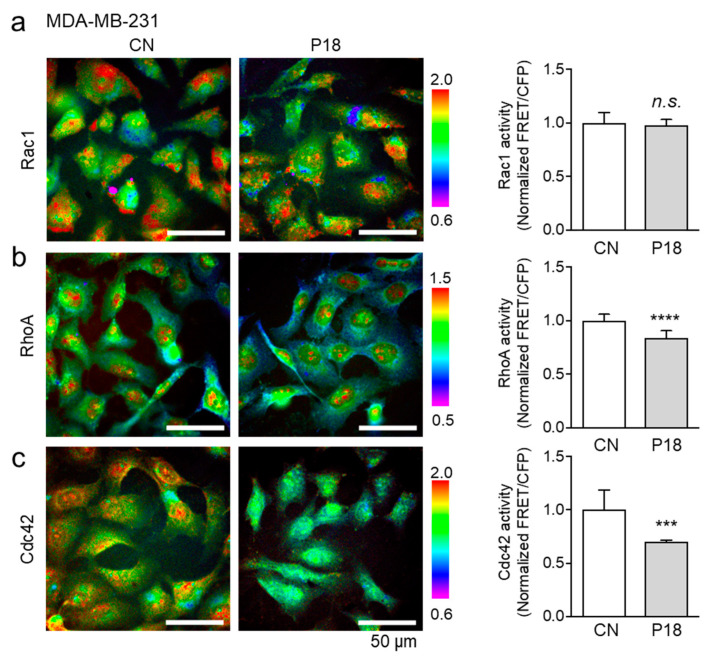
Effects of P18 on GTPase activities of Rac1, RhoA, and Cdc42 in MDA-MB-231 breast cancer cells. FRET-based GTPase activity is color-coded in individual cells, in which data are normalized and compared to the control group (CN). The scale bars indicate 50 μm. *** *p* < 0.001, **** *p* < 0.0001, n.s.: no significance. (**a**) No detectable effect of P18 on Rac1 GTPase activity. (**b**,**c**) Suppression of GTPase activities of RhoA and Cdc42, respectively, in response to P18.

**Figure 7 cancers-16-02230-f007:**
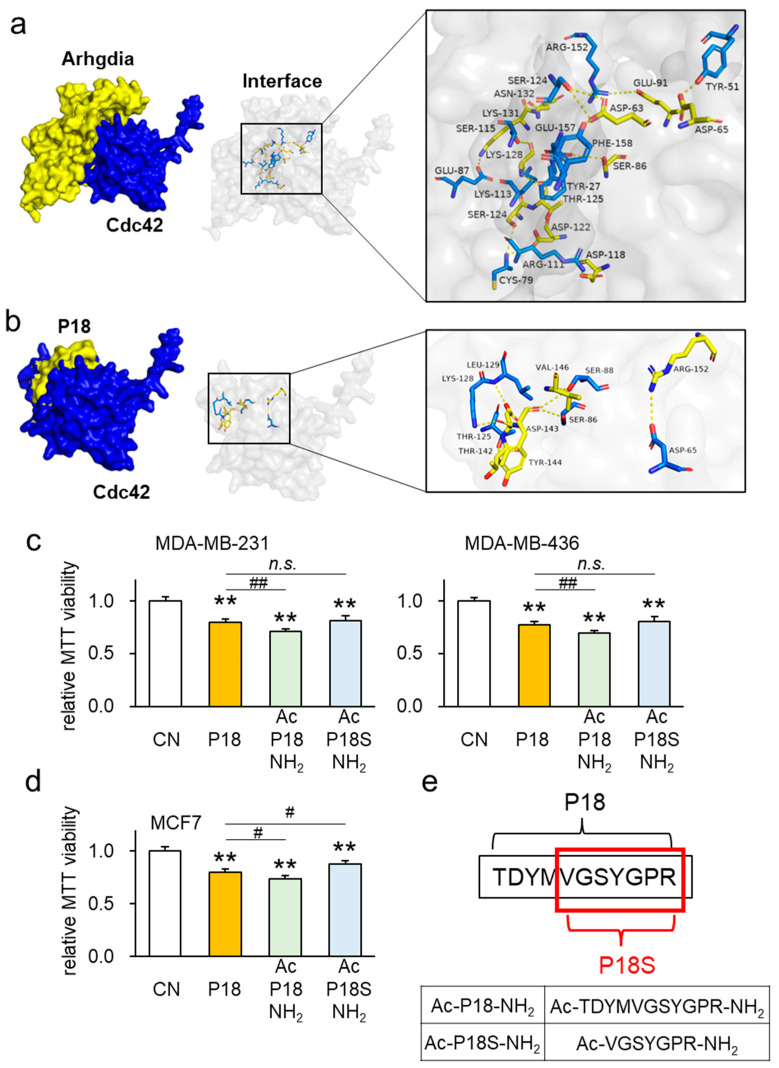
Evaluation of amino acid residues in P18 through molecular docking analysis and modification of both ends of P18. CN = control. The double asterisk indicates *p* < 0.01 for the comparison to CN. The single and double hash marks indicate *p* < 0.01 and 0.05 for the comparison to the P18 treatments. n.s.: no significance. (**a**) Predicted interactions between Cdc42 and Arhgdia (Cdc42 in blue, and Arhgdia in yellow). (**b**) Predicted interactions between Cdc42 and P18 (Cdc42 in blue, and P18 in yellow). (**c**,**d**) MTT-based viability of MDA-MB-231, MDA-MB-436, and MCF7 breast cancer cells in response to two modified P18 analogs. (**e**) The schematic diagram of modified P18 analogs.

**Figure 8 cancers-16-02230-f008:**
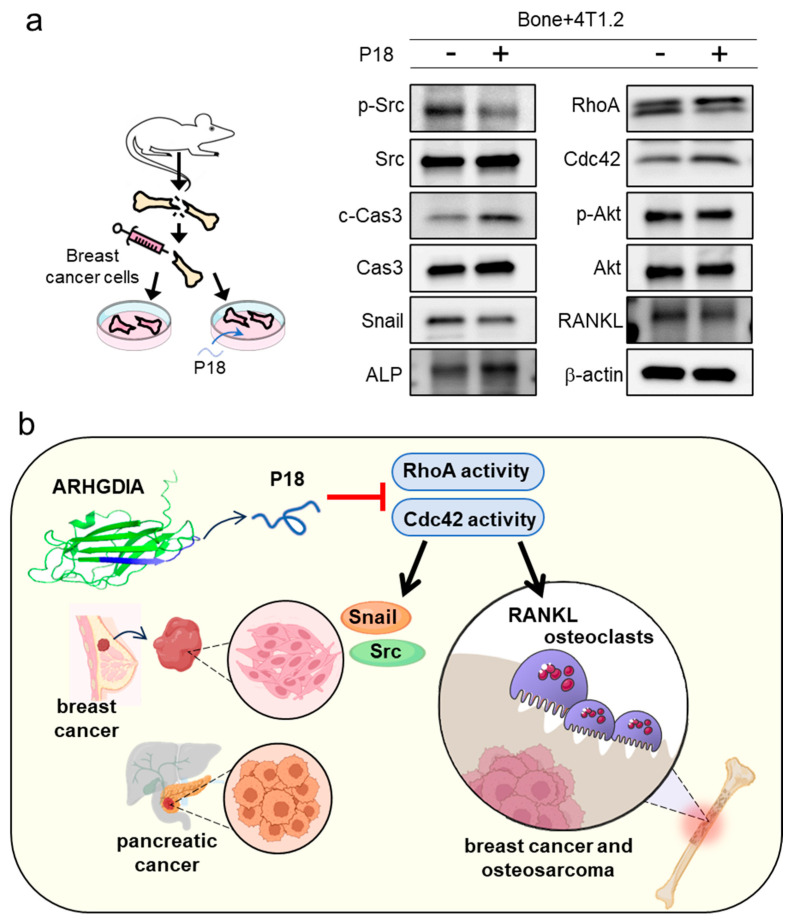
Osteoprotective effects and the proposed mechanism of P18 action. (**a**) Reduction in p-Src and Snail, and an increase in cleaved caspase 3 (c-Cas3) and alkaline phosphatase (ALP) by the daily administration of 25 µg/mL of P18 in ex vivo bone culture. The levels of RhoA, p-Akt, and RANKL are slightly reduced. However, the level of Cdc42 is slightly elevated. The uncropped blots are shown in Appendix A. (**b**) Proposed mechanism of antitumor and osteoprotective action of P18.

## Data Availability

The data presented in this study are available in this article (and Appendix A).

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
