# Peer review of "P18: Novel Anticancer Peptide from Induced Tumor-Suppressing Cells Targeting Breast Cancer and Bone Metastasis"

_cancers, 2024, doi:10.3390/cancers16122230_

Round 1
Reviewer 1 Report
Comments and Suggestions for Authors
The authors of this study entitled "P18: Novel Anticancer Peptide from Induced Tumor-Suppressing Cells Targeting Breast Cancer and Bone Metastasis" describe the development and testing of the P18 peptide against breast cancer cells. Although the study is interesting, several points should be addressed:
1. Apart from the effect of P18 on mesenchymal stem cells, in order to firmly conclude that P18 exerts no effect on normal cells, the authors ought to test the effect of P18 on normal or normal-like cells like MCF10A or other similar models.
2. The authors show that P18 affects Cdc42 and RhoA activity. Since P18 is added extracellularly how the peptide affects intracellular protein activity is not clear and should be investigated. For example, is P18 endocytosed or binds cell surface receptors?
3. The direct interaction between P18 and Cdc42 is shown by protein-protein docking. The interaction should be verified with immunoprecipitations from cell extracts, especially given the fact that P18 is added extracellularly (related to comment #2).
4. To support the notion that P18 exerts osteoprotective functions more experiments are needed, like in vitro osteoclast-osteoblast differentiation assays. This is critical since the western blots at figure 8 originate from bone tissue inoculated with cancer cells and therefore it is unclear if and how P18 affects bone osteoblasts/osteoclasts themselves.
5. In the discussion the authors mention that P18 is not expected to cause unwanted immune responses. P18 originates from an intracellular protein secreted in the microenvironment and there has been multiple documented cases where extracellular presence of intracellular proteins induce immune responses. The authors should add more arguments supporting this speculation.
Comments on the Quality of English LanguageN/A
Author Response
Dear Cancers editorial office,
We are pleased to submit the revised manuscript, entitled "P18: Novel Anticancer Peptide from Induced Tumor-Suppressing Cells Targeting Breast Cancer and Bone Metastasis". Thank you for the editorial assistance and valuable comments and suggestions. To revise the manuscript, we conducted two additional experiments. We appreciate the opportunity to re-submit the manuscript. Here, we have prepared point-by-point responses in this letter and have included a revised article, with the revised parts highlighted in red.

Reviewer 2 Report
Comments and Suggestions for Authors
The article presenting new peptides with anti-cancer activity is interesting. It contains many methods and examines the mechanisms of their action. There are a few shortcomings and comments that could improve the entire manuscript.
2.3 MTT assay – why the cells were treated for 2 days and MTT was done post 4 days?
EdU assay – how long were the cells incubated with examined peptides?
2.8 what is CM?
In the discussion, it is good to avoid words such as: they dramatically reduce the viability of cisplatin and taxol. It is rather a modulatory effect. The line is lower in the combination group than in the chemotherapeutic alone group for the entire length by a similar amount. IC50 is 2x lower in the combination of peptide and drug than in the drug alone -this is true but no other result confirms it. The limits of these results should also be indicated. It would be good for the authors to discuss whether any peptides have therapeutic use in the clinic. What would be the route of administration - in the form of infusion? Or continuous infusion? Can we predict the stability of such a peptide? These elements were missing from the discussion and it is good to include them.
Comments on the Quality of English LanguageEnglish is good
Author Response
Dear Cancers editorial office,
We are pleased to submit the revised manuscript, entitled "P18: Novel Anticancer Peptide from Induced Tumor-Suppressing Cells Targeting Breast Cancer and Bone Metastasis." Thank you for the editorial assistance and valuable comments and suggestions. To revise the manuscript, we conducted two additional experiments. We appreciate the opportunity to re-submit the manuscript. Here, we have prepared point-by-point responses in this letter and have included a revised article, with the revised parts highlighted in red.

Round 2
Reviewer 1 Report
Comments and Suggestions for Authors
The authors addressed all the comments. Therefore, I recommend the paper to be accepted to its present form.